# Synergetic Relationship between Urban and Rural Water Poverty: Evidence from Northwest China

**DOI:** 10.3390/ijerph16091647

**Published:** 2019-05-11

**Authors:** Wenxin Liu, Minjuan Zhao, Yu Cai, Rui Wang, Weinan Lu

**Affiliations:** College of Economics and Management, Northwest A&F University, Yangling, YL 712100, China; liuwenxin@nwafu.edu.cn (W.L.); caiyu1785@nwafu.edu.cn (Y.C.); xiaoxue@nwafu.edu.cn (R.W.); luweinan@nwafu.edu.cn (W.L.)

**Keywords:** urban and rural water poverty, integrated weight, synergistic theory, northwest China, water resources management

## Abstract

Combining the measurement of urban and rural areas to link water and poverty provides a new insight into the fields of water resources management and poverty alleviation. Owing to rapid urban development, water resource conflicts between urban and rural areas are gettingbecoming more intensified and more complex. This study details the application of a water poverty index (WPI) using 26 indicators to evaluate urban and rural water poverty in northwest China during the period 2000–2017. This study also analyzes temporal variations of urban and rural water poverty by the kernel density estimation (KDE). We found that the level of water poverty is gradually declining over time and the improvements in urban and rural areas are not harmonious. Additionally, it applies the synergic theory to analyze the relationships between urban and rural water poverty. The correspondence analysis between urban and rural water poverty is significant because of the synergic level results. The results show that there are four primary types in northwest China: synchronous areas, urban-priority areas, rural-priority areas, and conflict areas, and their evolution stages. The results suggest the need for location-specific policy interventions. Furthermore, we put forward corresponding countermeasures. The research findings also provide a theoretical foundation for the evaluation of urban and rural water poverty, and a regional strategy to relieve conflict between urban and rural water poverty.

## 1. Introduction

Availability of water is the primary contributor to a country or region’s economic growth, and is a major constraint to the development of all countries with low water access, especially those located in arid and semiarid areas [1]. There is a strong understanding that water is one of the most stressed resources, and it is playing an increasingly important role in poverty alleviation and economic development in the world [2]. Water shortage is both a cause and a consequence of poverty [3]. In most areas of the world, water is central to poverty [4]. Rapid population growth increases the domestic and agricultural demand for water resources. At the global scale, agriculture accounts for over 70% of all water withdrawn by the municipal, industrial (including energy) and agricultural sectors, and water shortage decreases agricultural productivity, affecting both incomes and food security [5]. Improper planning and management can mean that an activity will be unregulated, and can lead to a range of negative social, economic and environmental impacts [6]. The low or lack of access to safe water which results directly or indirectly results in decreasing human productivity and living quality [7]. The resulting water resource shortage limits further development [8]. Hence, its provision is central to poverty alleviation, which is why this paper begins with a brief discussion of how water and poverty are interconnected.

The relationship between water and poverty has become a growing concern for governments and scholars. An influential theory system has been created by Sullivan, who put forward the relationship between water and poverty through the Water Poverty Index (WPI). The index also reflects social and economic factors, especially those related to water resources and water supply facilities, capacity, efficiency, and environmental quality [9,10]. Using this framework can provide a theoretical basis for integrated water resources management, and through integrated management, achieve the goal of human-water harmonious coexistence [11]. So far, the WPI has evaluated the water resources situation at multiple scales, including the urban [12,13], rural [14,15], basin [16,17,18], national [19], province [11,20], county [4], town [13], and community [21,22] scale, mainly related to water availability, accessibility, quality, environmental impact, and social and economic factors [10,23]. Although scholars agree on the advantages of the WPI [22,24], there are some weaknesses that must be addressed. First, the existing research mainly uses the single weight method-equal weight [9,10,24] or different weight model [1,23,25,26,27] to evaluate water poverty. In recent years, the popularity of composite indicators to measure the situation of water poverty has increased [28]. However, this process of weight is less discussed and is marked by certain controversies, especially its current lack of explanation of its weight determination method [29]. Equal weight cannot sufficiently reflect subjective and objective conditions of different indicators, and different weights may ignore the homogeneity of the components [20]. Second, existing research is focused on specific times, namely a single year; however, it has ignored the panel data [29]. Panel data will help to find the evolutionary mechanism of water poverty and it is possible to obtain accurate results with this data, which will have a beneficial effect on policymaking [20]. Third, a further limitation is that scholars have focused on water poverty in rural or urban areas separately [24,30], rather than analyzed urban and rural areas jointly. Water poverty should not only be evaluated according to the integrity of the index system, but also according to the correlation between scales. Water poverty is a problem concerned with multiple systems such as water resources, economy, society and environment [16]; however, the water poverty index measures at a single scale, and current study is limited to interactive coupling of water poverty and economic poverty [31]. At the same time, the research perspective also focuses too much on the evaluation of rural areas or water resources shortage, while ignoring the “crowding out effect” of urban areas on rural areas, severing the link between urban and rural areas, which is bound to lead to the failure of policies, and is not conducive to the improvement of regional water resources shortage [32]. To this end, only through knowing the exact the urban and rural development potential of spatial distribution features of urban and rural water resources shortage is it possible to realize the rational allocation and effective utilization of water resources [33].

This study analyzes water poverty issues in northwest China (semi-arid and arid regions) in this context. China faces a water shortage. The valaue for per capita water resources is only 2100 m^3^, which is less than a quarter of the world’s average, and there is an uneven distribution of water resources on the national scale [34]. From a regional perspective, these water shortages are expected to worsen, especially in the northwest arid and semi-arid areas [35]. Moreover, the water policy of the Chinese government considered “the urban first” and “the industrial first” for an extended period. This led to a binary structure, unconducive to coordinated societal development [36]. In its allocation of water resources, China favors urban areas, as it prioritizes industrial development to boost the national economy. However, an industrial structure that encourages high water consumption, severe pollution, and a low efficiency is unsustainable; thus, irrigation water for the agricultural sector is not guaranteed. Moreover, China’s low agricultural productivity creates a water dependence that directly affects the stability of rural areas [37]. The development of urban and rural water resources is facing a serious imbalance. The construction and management of urban water resources is relatively perfect, but the construction and management of rural water resources is seriously lagging behind, and cannot even meet the reasonable needs of rural residents [38]. This has seriously affected the improvement of rural residents’ right to survival and development, and hindered the improvement of rural residents’ ability. Insufficient water supply in rural areas has become a bottleneck, and has restricted rural development and the improvement of rural social productivity [37]. Indeed, when water resource deployment is close to exceeding the limit for rural areas, urban areas must stop using rural water to prevent deterioration in the rural ecological environment. Hence, although it can accelerate urban socio-economic development, the recovery of rural water resources is uncertain [39]. Thus, reasonable use of limited water resources and their optimal allocation between urban and rural areas are critical.

The aim of this paper, therefore, is to evaluate urban and rural water poverty in Northwest China in terms of the integrated weight methods. In addition, this paper applies to provide a platform for policy-makers through the comprehensive assessment of regional water poverty, more insight into the problem of water resource shortage can be gained by taking a comprehensive approach that analyzes the spatial and temporal variability in the water resources condition. It will appeal to all stakeholders and will support effective water resource management and the achievement of commitments to reducing water poverty (James and Dermot, 2004). Therefore, this paper reveal break past limitations that arise in simplified analyses using a single method, as well as formulate more reasonable regional sustainable development policies. This article is useful for water resources studies in developing countries. The remainder of this paper is organized as follows. Section 2 introduces the study area. Section 3 describes the structures of the WPI, KDE, synergistic theory, and constructs a synergistic mechanism between urban and rural areas. Section 4 presents the WPI scores and urban-rural water poverty condition and discusses the main findings. Finally, Section 5 provides conclusions and policy recommendations.

## 2. Water Issues in the Study Area

The northwest region encompasses Shaanxi, Gansu, Qinghai, Ningxia, and Xinjiang provinces (Figure 1). The northwest region has an arid and semi-arid climate. It is dry and significant water shortages prevail. As the area with the most serious water shortage, the total annual water resources in the northwest arid region were 25.31 billion cubic meters in 2017 [34]. After deducting the water resources that are difficult to use or cannot be used, the actual figure for the per capita water resources in the northwest region was about 990 m^3^, which is less than 1/10 of the world’s average [40]. As the largest agricultural water demand area, the annual average rainfall is below 200mm, and the annual evaporation is as high as 1000–2800 mm [41]. The rainfall is less than the water demand of farmland crops, which makes the ecological water demand of the northwest region higher than that of other regions in China [42]. As the poorest region in China, it is home to a poor population of nearly 102 million people, with nearly 10 million people live in poverty, mostly dependent on agriculture-based livelihoods [43]. As the most vulnerable area of the ecological environment, it covers an area of 3.0344 million km^3^ and accounts for 31.6 percent of the country’s total area, of which 1.606 million km^3^ were undergoing desertification in 2017 [43]. The fragile ecological environment, fewer per capita water resources, large ecological water demand and severe poverty are interwoven together, which makes it an arduous task to improve water shortage and alleviate poverty in northwest China [44].

Currently, the urban and rural areas in northwest China are faced with major problems. Firstly, they have a low water efficiency. Agricultural water consumption accounts for over 90% of the total water consumption in the northwest China. However, agricultural water use efficiency is less than 40%, compared with over 80% in developed countries during the same period [44]. Secondly, rural water supply has been severely squeezed by urban needs. The quality of domestic water is an important indicator of the civilization level of a country or a region. However, the current situation is that the domestic water consumption of rural areas is poor in northwest China. By the end of 2017, the rural water population in northwest China accounted for less than 70% of the rural population [34]. Poor people are scattered and live in areas with poor water conservancy conditions and difficulty in drinking water and livestock. At the same time, the water shortages, the construction of water conservancy infrastructure and the imperfection of water use system aggravate the transfer of rural population to urban areas, leading to the increasingly serious phenomenon of rural hollowing out [44]. Thirdly, rural areas are faced with serious water pollution. With the development of urbanization, the urban population increases rapidly, and it transfers many polluting industries to rural areas, resulting in the rapid increase of rural sewage [45]. However, most of the financial investment in pollution control is transferred to urban areas. The treatment of water pollution in rural areas is on the whole deteriorating, and the fragile rural ecological environment is facing great challenges. Fourthly, urban and rural contradiction in water use for industry and agriculture intensified. Due to perceived preferential government policies that favor urban and industrial water users, conflicts tend to arise between urban and rural water users [11]. Fifthly, water resources management and planning are irrational. Restricted by the ecological environment and water resources, many problems have also appeared in the urbanization process in northwest China, such as relying on industrial and mining resources construction, making the urban development scale the priority rather than the local water supply, and water difficulties. This is closely related to the government intervention and unreasonable planning, while at the same time, excessive urbanization and a lack of reasonable water areas are closely related to the industry and urban layout planning [46].

Therefore, with the background of increasing water crisis, the reasonable allocation of water resources between urban and rural areas has become an urgent problem to be solved in economic development and ecological environment improvement in northwest China. This also plays a crucial role for sustainable development in the future.

## 3. Methods

### 3.1. Water Poverty Index, and Its Indicators

The urban and rural water poverty composite system is complex. It includes social and economic resources, the ecological environment, and various other elements. However, the most widely used indicator is the WPI. The methodology of this study is based on the WPI model [9,10,24], which evaluates the extent of water shortage through five components: Resources, Access, Capacity, Use, and the Environment, as follows:(1)$$WPI=w_{r}\times Resource+w_{a}\times Access+w_{c}\times Capacity+w_{u}\times Use+w_{e}\times Environment$$
$w_{i}$ are the weighting factors.

The choice of indicators (Table 1) within these five aspects of WPI model is not simply drawn from the water poverty routine index system but reflects the specific situation in northwest China.

Resources. Indicators for Resources consider the physical availability and reliability of water resources in the chosen study area [10]. They should reflect water availability and emphasize the comparative advantage given by available water resources, and whether there is population pressure on available water resources in the study area [47,48].

Access. Indicators for Access consider the extent to which people have access to agricultural, non-agricultural and domestic water use in the region [48]. They reflect not only the distance to a safe source, but also population with reasonable access to an adequate amount of safe drinking water and sanitation for better health and well being [6]. In the urban and rural areas, the indicators should reflect the significance of adequate and safe access to industrial water and agricultural water which leads to a decreased amount of time spent on water collection and effluent discharge.

Capacity. This component exhibits the effectiveness of people’s ability to manage water. With a close relationship in between society and water management, the importance of social and economic capacities management of water scarcity is increasingly being recognized. It reflects a social adaptation to water shortages [24]. The indicators should reflect the ability to improve water use efficiency and water resources management, the ability in the face of water conflict, water pollution and water press, and the ability to read, have access to information, understand water-related issues and, in some ways, think and act to manage water [29].

Use. Regarding use, the indicator correlates with the ways in which water is used for different purposes [22] and its contribution to the wider economy, because water use is an essential pre-requisite for human activity. Water consumption tends to increase with economic development [44]. The indicator should reflect efficiency of usage of available agricultural water. Domestic and industrial uses are two major water uses that are considered as indicators of water availability.

Environment. Regarding the environment, this indicator measures environmental factors influencing the quality and quantity of agricultural, industrial and domestic water [17]. Maintaining the quality of environmental and ecosystem health is important for achieving sustainable use of water resources. Environmental components are applied directly in the local ecological environment, and indirectly reflect the environmental impact of humans on the ecosystem and the variable degree of influence degree on water resources. The indicators reveal the pressure of human activities on the environment from the agricultural, industrial and domestic sector. Additionally, they address how do humans deal with these pressures [22].

### 3.2. Kernel Density Estimation

In parametric regression analysis, it is assumed that the data distribution conforms to a certain specific behavior, such as linearity, linearity or exponential behavior, etc., and then specific solutions are sought in the target function family, that is, unknown parameters in the regression model are determined. However, experience and theory show that there is often a big gap between this basic assumption of parametric model and the actual physical model, and these methods cannot always achieve satisfactory results. Compared with parameter estimation, kernel density estimation method does not use prior knowledge of data distribution and does not attach any assumptions to data distribution. It is only a method to study the data distribution characteristics from the data sample itself. Therefore, in the statistical theory and application fields are highly valued [49]. This study uses Eviews 6.0 software, to compute the kernel density distribution of water poverty in China. Kernel density estimation (KDE) is usually used to describe distribution of economic movement, it can identify variation of differences through dynamic and intuitive map [50]. In this study, the KDE can effectively reflect all yearly the dynamic movement trend of water poverty in China. For data *x*_1_, *x*_2_, …, *x_n_*, the formula is as follows:(2)$$\hat{f_{n}}\left(x\right)=\frac{1}{nh}\sum_{i=1}^{n}K\left(\frac{x-x_{i}}{h}\right)$$
where *K* is the kernel function, h is the window width, and n is the sample size.

Kernel functions are weighted functions, including Gaussian, Epanechnikov kernel, triangle kernel, and four times kernel, selected based on the data. In this study, we make estimates by using the Gaussian kernel function:(3)$$\mathrm{Gaussian}:\frac{1}{\sqrt{2\pi }}e^{-\frac{1}{2}t^{2}}$$

### 3.3. Synergistic Theory

The synergistic theory is a comprehensive theoretical framework based on an interdisciplinary approach. A composite system refers to the cooperative co-evolution of composite systems and environmental material, energy, and information exchange, after birth, growth, maturity, recession, and the evolutionary process of death, eventually achieving a certain level of equilibrium development. Its evolution path is in line with the S-type curve. In the same vein, the logistic growth model describes the process of a composite system in terms of synergy evolution; its equation assumes that, at time *t* and for a compound system change, Alpha is the multiplication factor for the composite system:(4)$$\frac{dX}{dt}=\alpha X\left(1-X\right)$$

The right-hand side is the time growth factor, *X*, a dynamic factor, and (1 − *X*) is a reduction factor; this is equal to its amount over time, as the development of the composite system evolution mechanism is nonlinear for positive and negative feedback mechanisms [51].

China’s urban and rural water poverty composite system includes two subsystems, namely urban water poverty and rural water poverty, denoted as *X* = *X* (*t*) and *Y* = *Y* (*t*), respectively; time, *t*, is continuous and differentiable. According to the system dynamics method, the interaction of the two-line system dynamics model is expressed by the following logistic model:(5)$$\frac{dX}{dt}=f_{1}\left(X,Y\right)=\frac{r_{1}X\left(N_{1}-X-\alpha_{1}Y+\beta_{1}Y\right)}{N_{1}},$$
(6)$$dY/dt=f_{2}\left(X,Y\right)=r_{2}Y\left(N_{2}-Y-\alpha_{2}X+\beta_{2}X\right)/N_{2},$$
(7)$$\alpha_{1}+\beta_{1}=\alpha_{2}+\beta_{2}=1$$
where *N*_1_ and *N*_2_ represent the development limit of the urban and rural water poverty subsystems, respectively, with *N*_1_ = *N*_2_ = 1; *r*_1_ and *r*_2_ are the adaptive rates of the two subsystems’, and *α* and *β* are their cooperation and competition coefficients, respectively [52]. For example, *α*_1_ is the inhibiting effect of urban subsystem development on the rural subsystem, and *β*_1_ promotes this effect. Hence, the model reflects that urban and rural water poverty, which both restrain and promote each other (i.e., they, co-evolve).

The conditions f_1_ (*X*, *Y*) = 0, f_2_ (*X*, *Y*) = 0, *α*_1_ − *β*_1_ = a_1_, and *α*_1_ − *β*_1_ = a_2_ are set to obtain four stationary-state solutions: A1 (0, 0), A2 (1, 0), A3 (0, 1), and A4 ($\frac{1-a_{1}}{1-a_{1}a_{2}}$,$\frac{1-a_{2}}{1-a_{1}a_{2}}$). However, the system is not stable in A1 (0, 0). The steady-state solutions for A2 (1, 0) and A3 (0, 1) correspond to the demise of the state of the system’s *X* and *Y*, respectively, and these three types of stationary-state solutions are not sustainable. The complex co-evolution of urban and rural water poverty should undergo steady state solutions analysis for A4 [53]. According to the size of the parameters and degree of synergy evolution, this co-evolution state of urban and rural areas is divided into four categories: synchronous, urban-priority, rural-priority, and conflicting relationships (Table 2).

The conflicting type refers to the fact that the development of space, resource allocation, industrial policy, and rural water pollution damage the structure and function of the urban water resource system, with rural water resources being used to the exclusive advantage of urban users. On the contrary, a synchronous relationship reflects the interaction and economic complementarity of urban and rural water poverty, gradually improving the coordination mechanism of the resource and environment. Specifically, for the urban water system to support the rural water system, the latter should provide conditions that enable the common development of both systems. However, the urban priority and rural priority typically struggle between conflicting and synchronous types.

According to the different types and parameter sizes of the co-evolution of urban-rural composite systems, the urban-rural composite system was divided into four evolutionary stages (Table 3) by referring to the related researches on coordinated development and coupling coordination [53]. The traditional approach to estimating the optimization of model parameters, such as the least squares method and maximum likelihood method, the resulting parameter estimates are typically local rather than global optimal solutions. However, Equations (5)–(7) are nonlinear equations, satisfactory accuracy is often difficult to achieve with the above solutions when solving such problems, and the maximum likelihood method needs to know the exact distribution of parameters. Hence, this paper adopts the Genetic Algorithm (GA) to solve model parameters. GA is a useful method for dealing with complex optimization problems which simulates the rules of survival of the fittest and the mechanism of chromosome information exchange within a population in the process of biological evolution. Its basic principle is based on several generations of superposition, to get the optimal calculation results. It can be used for multiple search points and only uses the value of the objective function to search. It has the advantages of high search efficiency, flexible method, low requirement for the objective function and fast calculation speed [54]. There are many forms about GA. Simple Genetic Algorithm (SGA) is adopted in this paper, which has the characteristics of strong applicability, robust global optimization, less computation and high precision of solution, and concise algorithm control parameter setting technology, and has been widely used in various optimization fields. It is easy to write a simple general algorithm. In application, only the definition of specific objective function and the setting of GA algorithm control parameters need to be modified [55]. Based on these characteristics, SGA has been widely used in various optimization fields.

Let the general optimization problem be,
(8)$$\{\begin{array}{c} \min f\left(c_{1},c_{2},\ldots ,c_{p}\right)\\ a_{j}\leq c_{j}\leq b_{j} \end{array}\;\;\;\;\;\;\;\;\;\;j=1,2,\ldots ,p$$
where, $\left\{c_{j}\right\}$ is p variables, $\left\{a_{j},b_{j}\right\}$ is the initial change interval of $\left\{c_{j}\right\}$; f is a nonnegative optimization criterion function. According to the optimization performance of each operator of the standard genetic algorithm, the variable change space of excellent individuals generated by the first two evolutionary iterations is used as the new initial change interval of the variable, and the algorithm enters the discrete coding of the initial change space of the variable. Thus, the cycle is accelerated until the optimization criterion function value of the optimal individual is less than a set value or the algorithm runs for a predetermined number of accelerated cycles, and then the optimal individual or an excellent individual in the current group is then designated as the SGA result. The calculation principle of SGA and the specific implementation process of the model are mainly referred to Jin, J.L. et al., 2001 in this paper [55]. As the process is very complex, it is unnecessary to elaborate here.

### 3.4. Assigning Weights to the Indicators

Weights are commonly used in multidimensional indexes and these are determined by each dimension value to obtain a comprehensive measure. However, the choice of weight is a key problem, and simple weighting methods ignore the inherent relationships between indicators [56]. The assignment of weights also influences the reliability and accuracy of the results, thus influencing decision-makers [20]. To achieve the most reliable results, this study applies analytic hierarchy process (AHP) and Principal Component Analysis (PCA). AHP synthesizes uncertainty, it is based on expert judgments, and it plays a significant role in the relative importance of the evaluation as well as the analysis indicators [29]. However, while they reflect the specific situation of the indicators, they do not reflect their economic and technical significance [57,58]. Hence, we introduced PCA to offset this disadvantage. PCAs are based on the analysis of measurable data, but may result in variations in their weights’ importance [59]. PCA is used to replace the original indicators with a new set of unrelated comprehensive indicators by recombining the original indicators which have a certain correlation, and the new comprehensive indicators can contain most of the information of the original indicators. It can eliminate the correlation between evaluation indexes and help to describe the relative status of samples more objectively. Therefore, to assign indicator weights systematically, this study combines AHP and PCA. In the process of to ascertain the integrated weights, the importance of the AHP and PCA were 0.5. These integrated weights are then assigned to the variables to highlight each indicator’s importance and accuracy [58]. This is suitable when complementary and uncertain information is merged, thereby allowing subjective and complex information to be transformed into deterministic decisions [60]. Due to the limited space, we have not listed the specific formula and calculation process of weight [1,20,25]. In addition, the AHP-determined subjective weighting vector is defined as Table A1, Table A2, Table A3, Table A4, Table A5, Table A6, Table A7, Table A8, Table A9, Table A10, Table A11 and Table A12 (Appendix A). The results are presented in Table 4.

### 3.5. Symbiosis Mechanism of Urban and Rural Areas

The development of an urban water resources system provides conditions for the improvement of rural water resources system (Figure 2). The development of the rural water resources system provides support for the development of the urban water resources system, forming mutual promotion between urban and rural systems. The two systems compete in terms of space and opportunity, and the over-development of water resources and the contradiction between urban and rural water resources are intertwined, resulting in the mutual suppression of the two systems. The co-evolution of the urban and rural systems refers to the structure and process formed in the interaction between different urban and rural elements in the long-term evolutionary interaction, driven by the inhibitory and promoting effects. From the perspective of biological evolution, the most basic coevolutionary mechanism of an ecosystem is competition and cooperation, which is also an important driving force for the evolution and development of urban and rural water resources systems. According to the synergetic theory and water poverty theory, the internal factors of urban and rural water poverty can be systematically divided into five units of resources, facilities, capabilities, use and environment, which constitute the symbiosis of the urban and rural complex system. Each symbiont and each element of the symbiont stimulate the overall vitality of the system through moderate competition and promote upgrading of the system. In addition, the reciprocity between the symbionts and their elements is conducive to the optimal allocation of resources and the expansion of each other’s development space. However, the vicious competition will compress each other’s development space, causing the system to be unable to achieve its ideal development condition. Under the joint action of the internal mechanism and external policy, the system will go through the dynamic evolution process from germination, formation, growth and maturity of the collaborative evolution between systems.

## 4. Results and Analysis

### 4.1. Urban and Rural Water Poverty Changes in Northwest China

The estimation results show the actual urban and rural water poverty situation in northwest China from 2000 to 2017 (Table A13 and Figure 3). The results show that water poverty values vary widely from 0.118 to 0.443 in urban areas and from 0.146 to 0.352 in rural areas. The higher the score, the better the situation, the less water poverty. In Appendix A, the value change reveals the improvement degree between urban and rural water poverty and shows two main results: (i) the water poverty situation across northwest China is gradually improving, and (ii) the absolute difference in the improvement degree for urban and rural water poverty has gradually widened. These two results imply that the improvements in urban and rural water poverty have not been harmonious. The situation of water poverty in rural areas changes slowly, and the change in speed and degree is much lower than urban areas. This finding shows that urban areas, which are characterized by rapid economic growth, display accelerated improvement in water poverty.

To accurately analyze the temporal variation of water poverty in northwest China, we introduced KDE to analyze water poverty urban water poverty and rural water poverty respectively. According to the results, we observed temporal evolution of the water poverty, including the first year (2000), the middle year (2008) and the end year (2017). According to calculated results, we produced an urban and rural water poverty kernel density distribution. In Appendix A and Figure 3, obvious changes in the values reveal the improvement degree in the water poverty situation in urban and rural area. However, the differences are striking between urban and rural areas. Firstly, looking at the position of the estimated function in Figure 4, we see that from 2000–2017, the density distribution curve shifted to the right, reflecting the gradually improving water poverty situation, and the urban water poverty is improving faster than the rural area. Secondly, looking at the shape, the development level of water poverty is not strictly unimodal. The single peak of the urban area is more uniform, and the rural area exhibits single peak and double peak crisscross, so its development is extremely imbalanced. In the bimodal distribution in 2000, the first double peak is small and the second double peak is larger, indicating that the development of urban water poverty shows a trend of polarization, reflecting the period of rapid urban development and the contradictory demand for water resources. As time goes on, the distribution becomes unimodal in 2008 and 2017, showing that the polarization of urban water poverty had eased. Compared to urban areas, In the bimodal distribution in 2000, the right double peak of rural area is small, the water poverty gap between regions is not large. However, rural water poverty experienced two peaks in a row in 2008. This shows that the polarization of rural water poverty widened sharply during this period, and the water situation worsened. This may be the lagged effect caused by the development of the urban water resources system in the previous period, but the specific relationship between the two will be analyzed in the next part. In 2017, the twin peaks of rural water poverty decreased, indicating a decrease in polarization between regions. Thirdly, looking at the kurtosis, compared to the urban area, the shape of the rural water poverty distribution over 2000–2017 showed a tendency to develop to a broad peak form. Water poverty development presents obvious peaks in 2008; with advancing time, wave height obviously decreased, while the middle to right side area is gradually increasing; with peaks corresponding to the development level of water poverty, it shows that in many of the areas water poverty development levels have improved, despite the low level of growth in the region. It indicates that the low-level areas of water poverty development are more numerous than the high-level areas.

However, we have only analyzed the temporal changes of urban and rural water poverty, respectively, and ignored the spatial correlation between urban and rural areas in this section. The improvement of urban and rural water poverty may be based on a competition or cooperation mechanism, and the former can promote the coordinated development of urban and rural water poverty. This is further discussed in the next section.

### 4.2. Synergistic Types and Stages of Urban and Rural Areas in Northwest China

This study applies synergistic theory to analyze urban and rural water poverty in northwest China, uses a genetic algorithm to estimate the model parameters, and analyze the synergistic types and stages of the complex system (Table A14). Figure 4 shows that urban and rural areas that have poor water composite system evolution stages show significant regional differences; they can be divided into four types: synchronous areas, urban-priority areas, rural-priority areas, and conflicting areas. Compared to rural areas, most urban areas have different cooperation coefficients, showing significant imbalance in the cooperation intensity between rural and urban areas. It is worth noting that the absolute cooperation coefficient *a*_1_ between most urban areas and rural areas is not equal to the absolute cooperation coefficient *a*_2_ between the latter and the former, and the intensity of cooperation between urban and rural areas is also unbalanced. For instance, Xian’s urban to rural cooperation coefficient is 0.7033, whereas the rural to urban cooperation has a coefficient of 0.6181. Such a phenomenon is associated with different urban areas and causes rural areas to closely relate to their relative development level. The weak subsystem of Xian’s rural development relative to the urban area decreases the competition coefficient of urban subsystems relative to rural ones, thereby causing a “squeeze out” from the former to the latter.

Based on the co-evolution of the urban-rural system, the parameters were estimated by the GA algorithm, and the obtained values were used to explore and analyze the types and stages of urban and rural water poverty in northwest China. To clearly represent the synergistic relationship of urban and rural water poverty, 52 regions were divided into four categories. In Figure 5, 36 regions are in the stage of uncoordinated development, showing an obvious competition and contradiction. 16 regions are synergistic regions. However, 5 regions are in the infancy stage and have the highest water poverty score in urban areas. 11 regions were represented as rural-biased regions, 17 regions were represented as urban-biased regions, and 7 regions were represented as conflicting regions. Therefore, the coordination coefficient of urban and rural water poverty will reflect the urban water resources system construction and the rural water resources system construction in northwest China. The overall situation is not optimistic, almost 70% of the regions are still stay in the mutual restriction or isolated development stage. This unequal relationship between the increase of the urban and rural system will only improve the unilateral area, and does not produce a linkage effect, or feedback from the other party, and is unable to form effective collaborative development. In particular, the urban-priority regions that give priority to the development of the urban water resources system by squeezing rural water resources are occupy 1/3, which is consistent with the current reality of “urban squeezing rural” in China. The synergy coefficient indicates the types, stage and trend between urban and rural water poverty in northwest China, which plays an important role in the prediction of urban and rural development path and policy formulation in the future.

(1) Synchronous areas: These include Yanan, Ankang, Shangluo, Lanzhou, Jinchang, Jiuquan, Qingyang, Gannan, Yinchuan, Shizuishan, Wuzhong, Altay, Bortala, Kizilsu, Kashgar, and Hotan.

According to the dispersion of distribution points, Yanan, Ankang, Shangluo, Lanzhou, Jinchang, Jiuquan, Qingyang, Gannan, Yinchuan, Kizilsu, and Kashgar have good economic conditions and resource endowments, and the urban and rural water poverty presents a mutual benefit situation, so this paper will not present an in-depth discussion. Shizuishan, Wuzhong, Altay, Bortal, and Hotan are in the infancy stage. Limited by the economic and social basis and natural conditions, urban and rural water poverty develops relatively slowly, presenting a double low situation. In these areas, water resources and economic development are poorer than other areas, and the ecological environment has been severely damaged. Additionally, water saving is limited, for a large proportion of national agricultural production, and the water use efficiency is low.

(2) Rural-priority area: This includes Weinan, Yulin, Baiyin, Tianshui, Linxia, Xining, Guoluo, Turpan, Ili, Tacheng, and Bayangol. In this region, when the government increases the investment in the economy and water resources, it will only improve the rural water poverty situation, and the urban water poverty development will lag behind. For example, Weinan is an agricultural region, where agricultural water consumption is high. Its salient features are a dry windy climate and unevenly distributed rainfall. Owing to the water shortages, the ecosystem is fragile, leading to dry gas and animal husbandry being based on combined groundwater exploitation. Therefore, in order to maintain the stable development of rural areas, water resources and economic factors were given priority.

(3) Urban-priority area: This includes Tongchuan, Baoji, Xianyang, Hanzhong, Zhangye, Pingliang, Longnan, Guyuan, Haidong, Haibei, Hainan, Haixi, Urumqi, Karamay, Hami, Changji, and Aksu. The majority of the 17 regions show a formation or growth stage, and it is the development trend of urban preference. In these areas, increasing resource exploitation in urban areas will only unilaterally improve the urban water poverty, and it will not lead to rural water poverty improvement. For example, Tongchuan has a significant proportion of rural water resources that are used for urban development. This will ease urban water poverty, leading to the deterioration of rural water poverty. With economic and social development, water usage has surged. Owing to the rapid progress of industrial areas with different levels of shallow groundwater pollution, deep groundwater contains excessive fluoride. Moreover, the water use efficiency is low and wastage is widespread.

(4) Conflicting area: This includes Xian, Jiayuguan, Wuwei, Dingxi, Zhongwei, Huangnan, Yushu, and Shihezi. In these areas, the competition between urban and rural areas is unordered, and production factors, industrial policies and other aspects are prominent, while the development of urban water poverty leads to the destruction of rural water poverty. However, urban water systems are also subject to rural “reprisals”; for example, Xian’s water resources are low, but its economic level is quite high. Moreover, the social adaptation ability, water drainage facilities, government regulation, and water use efficiency are low. This creates a serious conflict between urban and rural water poverty.

## 5. Conclusions

In China, the government is the main department that distributes water resources. In recent years, the worsening water situation across society and intense competition between urban and rural water resources have meant that rural water shortages have become more serious than those in urban areas. Hence, conflicts between urban and rural water resource allocation increasingly tend to be the norm. Urban water resources system and rural water resources system should rise and fall together, and water shortage is no longer an absolute constraint, but the impact of urban and rural development. Biased attention on either side will not help solve the water shortage problem. Water shortage has become one of the most important factors restricting urban and rural development, especially in northwest China. Water shortage is influenced not only water supply and water demand, but also social, economic and geographical factors. Therefore, water shortage in northwest China is inevitable. However, it is no longer enough to evaluate the degree of water shortage to explain the complex water problems, but to comprehensively evaluate the probability, degree and social impact of water shortage.

In this context, this research showed that urban and rural water poverty is rising annually, while the value of complex systems of urban and rural development is increasing. It also demonstrated that urban water poverty improvement has accelerated rural water poverty in China. However, significant differences were found in the development of the two systems, with the urban water system showing greater development than the rural one. Moreover, about three-fifths of regions have serious urban and rural water conflicts.

The model in this study aimed to analyze the reasons behind these trends. The analysis of the complex co-evolution of urban and rural water poverty showed that these two relatively independent systems have different processes for checking and balancing the interactions between their internal mechanisms. Therefore, this study provided a new perspective of the mutual relations between urban and rural systems and added to research and development on urban–rural interactions by building a collaborative evolutionary dynamic based on the WPI and synergistic theory. Nonetheless, while the proposed model is novel, the urban and rural water poverty evaluation based on WPI and synergic theory remains a preliminary study. There are a few problems that have yet to be researched. Firstly, the different indicators selection criteria and weight method would produce different water poverty results, and further affect nonparametric estimation of distributions and synergic analysis. In the future, there is still a need to use a variety of other indicators, evaluation standard and other parametric methods, the analysis of the existing the robustness, and a reliability of inspection. Secondly, SGA is an evolutionary optimization algorithm; however, there are various limitations of the SGA despite many claims. In many complex nonlinear optimization models, it may not converge, or may converge to a local optimal. Substantial testing of the solution for different optimization parameters, different starting points may be necessary to ensure that it is reasonable to assume that a global optimal has been achieved. It maybe depend on the problem and input data. Thirdly, considering the effect of space, Exploratory Spatial Data Analysis (ESDA)—confirming the space data analysis method—could be a further interpretation of regional water shortage differences.

SGA is an evolutionary optimization algorithm; however, there are various limitations of the SGA despite many claims. In many complex nonlinear optimization models, it may not converge, or may converge to a local optimal. Substantial testing of the solution for different optimization parameters, different starting points etc. may be necessary to ensure that it is reasonable to assume that a Global optimal has been achieved. It maybe depend on the problem and input data.

## Figures and Tables

**Figure 1 ijerph-16-01647-f001:**
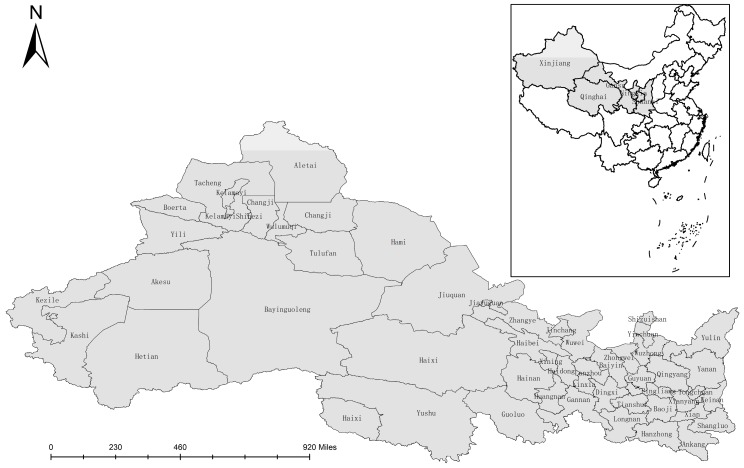
Study area in northwest China.

**Figure 2 ijerph-16-01647-f002:**
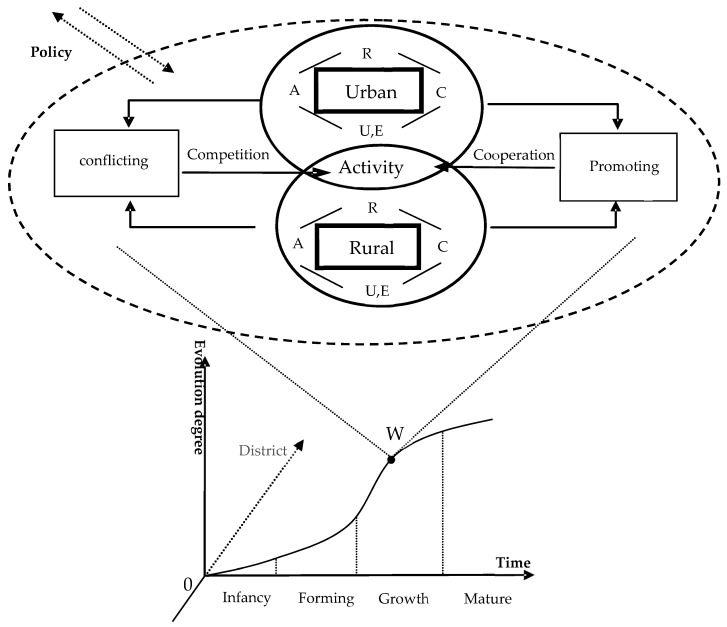
The synergetic evolution mechanism of urban-rural complex system.

**Figure 3 ijerph-16-01647-f003:**
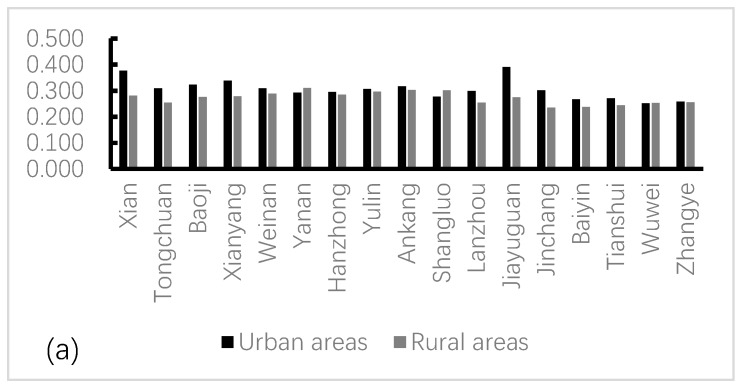
The urban and rural water poverty values (**a**,**b**,**c**) in northwest China.

**Figure 4 ijerph-16-01647-f004:**
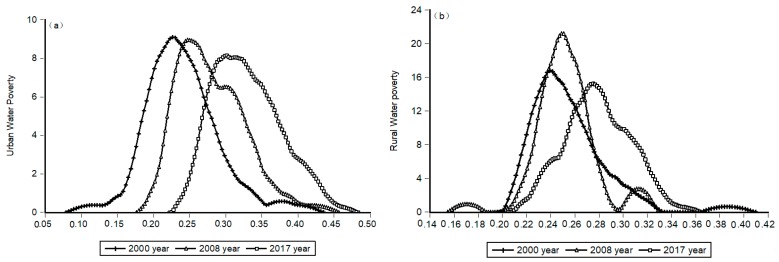
The kernel density distribution map of urban (**a**) and rural (**b**) water poverty in northwest China.

**Figure 5 ijerph-16-01647-f005:**
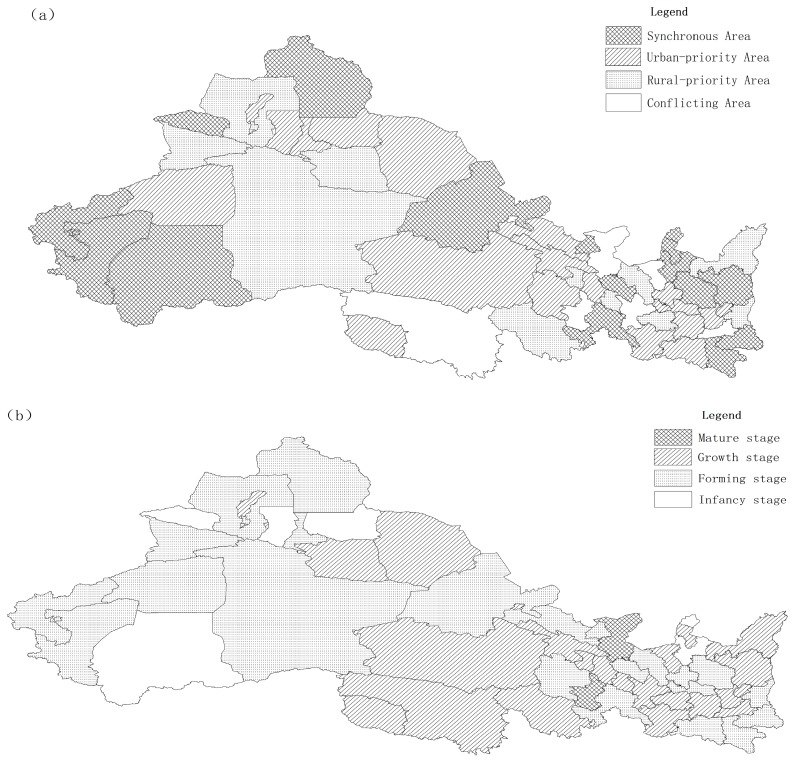
The symbiosis types (**a**) and stages (**b**) of urban and rural water poverty in northwest China.

**Table 1 ijerph-16-01647-t001:** Details of the WPI components, indicators, variables.

System	Component (Indicators)	Variable	Data Sources and References
Urban	Resources 1. Variability 2. Availability		
		(mm) variation of rainfall (+)	[4,34]
		(m^3^) Per capita water resources (+)	[9,10,34]
	Access		
	3. Supply	(%) Growth rate with access to clean water supply pipeline (+)	[11,34]
	4. Population	(%) Population with access to clean water (+)	[11,34]
	5. Sanitation	(%) Sewage treatment (+)	[20,34]
	Capacity		
	6. Economic	(CNY) Urban per capita income (+)	[20,43]
	7. Social	(%) Higher education enrolment rate (+)	[11,43]
	8. Government	(%) Financial self-sufficiency (+)	[22,43]
	Use		
	9. Domestic	(L) Urban per capita domestic water uses per day (+)	[9,10,43]
	10. Industrial	(m^3^) Industrial water use per 10,000 yuan (-)	[9,10,34]
	Environment		
	11. Stress	(m^3^) Volume of wastewater per 10,000 yuan (-)	[11,34]
	12. Quality	(m^2^) Per capita vegetation coverage (+) (m^3^) Sewage treatment (+)	[21,34] [11,34]
Rural	Resources		
	1. Variability	(mm) variation of rainfall (+)	[4,43]
	2. Availability	(m^3^) Per capita water resources (+)	[9,10,43]
	Access		
	3. Supply	(Km^2^) The actual irrigation capacity (+)	[22,43]
	4. Population	(%) Population with access to clean water	[1,43]
	5. Sanitation	(pc) Numbers of reservoir (+)	[22,43]
	Capacity		
	6. Economic	(CNY) Rural per capita income (+)	[20,43]
	7. Social	(%) Compulsory education enrolment rate	[11,43]
	8. Residents	(pc) Numbers of doctors per ten thousand people (+)	[22,43]
	Use		
	9. Domestic	(L) Rural per capita domestic water use per day (+)	[9,10,34]
	10. Agriculture	(m^3^) Agricultural water use per 10,000 yuan (+)	[9,10,34]
	Environment		
	11. Stress	(Kg) Chemical fertilizer use per hectare (-)	[22,34]
	12. Quality	(pc) Number of toilets per 10,000 people (+) (Km^2^) Soil and water loss control area (+)	[22,34] [11,34]

(+) means that the indicator is a positive value; (-) means that the indicator is a negative value.

**Table 2 ijerph-16-01647-t002:** The synergetic evolution classification of urban-rural complex systems.

Type	Parameter
Synchronous	a_1_ < 0, a_2_ < 0
Urban-priority	a_1_ < 0, a_2_ > 0
Rural-priority	a_1_ > 0, a_2_ < 0
Conflicting	a_1_ > 0, a_2_ > 0

**Table 3 ijerph-16-01647-t003:** The synergetic evolution period of urban-rural complex systems.

Type	Cooperation α	Competition β	Stages
1	0.75<α≤1	0<β≤0.25	Mature
2	0.5<α≤0.75	0.25<β≤0.5	Growth
3	0.25<α≤0.5	0.5<β≤0.75	Forming
4	0<α≤0.25	0.75<β≤1	Infancy

**Table 4 ijerph-16-01647-t004:** Weights of the WPI components, variables.

Component	Variable	AHP	PCA	Integrated
Resources (0.2)	variation of rainfall	0.0667	0.076	0.071
	Per capita water resources	0.1333	0.074	0.103
Access (0.2)	Growth rate with access to clean water supply pipeline	0.0622	0.092	0.077
	Population with access to clean water	0.0987	0.098	0.098
	Sewage treatment	0.0392	0.103	0.071
Capacity (0.2)	Urban per capita income	0.0825	0.063	0.073
	Higher education enrolment rate	0.0520	0.057	0.055
	Financial self-sufficiency	0.0655	0.078	0.072
Use (0.2)	Urban per capita domestic water use per day	0.1333	0. 105	0.119
	Industrial water use per 10,000 yuan	0.0667	0.079	0.073
Environment (0.2)	Volume of wastewater per 10,000 yuan	0.0622	0.068	0.065
	Per capita vegetation coverage	0.0392	0.089	0.064
	Sewage treatment	0.0987	0.017	0.058
Resources (0.2)	variation of rainfall	0.1333	0.053	0.093
	Per capita water resources	0.0667	0.116	0.092
Access (0.2)	The actual irrigation capacity	0.1056	0.032	0.069
	Population with access to clean water	0.0665	0.095	0.081
	Numbers of reservoir	0.0279	0.065	0.047
Capacity (0.2)	Rural per capita income	0.1000	0.052	0.076
	Elementary education enrolment rate	0.0500	0.113	0.081
	Numbers of doctors per ten thousand people	0.0500	0.086	0.068
Use (0.2)	Rural per capita domestic water use per day	0.0667	0.089	0.078
	Agricultural water use per 10,000 yuan	0.1333	0.069	0.101
Environment (0.2)	Chemical fertilizer use per hectare	0.0987	0.074	0.086
	Numbers of toilets per 10,000 people Soil and water loss control area	0.0392 0.0622	0.092 0.064	0.065 0.063

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
