# Peer review of "Synergetic Relationship between Urban and Rural Water Poverty: Evidence from Northwest China"

_ijerph, 2019, doi:10.3390/ijerph16091647_

Round 1

Reviewer 1 Report

Overall, this manuscript is well-written, and the topic is of great importance. I have the following major comments. In the methodology part, some methods are not clearly presented. For example, GA should be briefly introduced. Moreover, at least, the key equations to calculate the integrated weights in Table 4 should be listed. 

Other minor comments:

1. Line 14: kernel estimation -> kernel density estimation

2. The first paragraph of Introduction: a more recent paper, the study of Chen et al. (2016) can be cited to support the statements here.

Chen et al., 2016. Population, water, food, energy and dams. Renewable and Sustainable Energy Reviews, 56, 18-28.

3. Figure 1: the names are too small. Please revise.

4. Figure 3: the label of the Y-axis is missing.

5. Figure 4: the label of the X-axis is missing.

6. There are a few editing errors. I suggest the authors check the text carefully.

Author Response

Responses to Reviewer 1

AE1:  In the methodology part, some methods are not clearly presented. For example, GA should be briefly introduced. 

Response: Thank you for this helpful suggestion. Limited to words, this study don't have a detailed description of GA. But I have revised it.

It is that “The GA has the characteristics of a strong applicability, robust global optimization, small amount of calculation, high accuracy of the solution, concise algorithm control parameter setting technology and so on. Therefore, it is starting to be widely applied in various optimization fields.

 Let the general optimization problem be,

 ; f is a nonnegative optimization criterion function. According to the optimization performance of each operator of the standard genetic algorithm, the variable change space of excellent individuals generated by the first two evolutionary iterations is used as the new initial change interval of the variable, and the algorithm enters the discrete coding of the initial change space of the variable. Thus, the cycle is accelerated until the optimization criterion function value of the optimal individual is less than a set value or the algorithm runs for a predetermined number of accelerated cycles, and then the optimal individual or an excellent individual in the current group is then designated as the GA result. The calculation principle of GA and the specific implementation process of the model are mainly referred to Jin, J.L. et al., 2001 in this paper [54]. As the process is very complex, it is unnecessary to elaborate here.” (Line 285-Line 301, in ijerph-482170 - original.R1-Track Changes)

AE2: Moreover, at least, the key equations to calculate the integrated weights in Table 4 should be listed. 

Response: Thank you for this helpful suggestion. I have revised it. It is that “In the process of to ascertain the integrated weights, the importance of the AHP and PCA were 0.5.”. (Line 318-Line 319, in ijerph-482170 - original.R1-Track Changes). Namely, Integrated = (AHP+Entropy)/2

AE3: Line 14: kernel estimation -> kernel density estimation

Response: Thank you for this helpful suggestion. I have revised it.

AE4: The first paragraph of Introduction: a more recent paper, the study of Chen et al. (2016) can be cited to support the statements here. Chen et al., 2016. Population, water, food, energy and dams. Renewable and Sustainable Energy Reviews, 56, 18-28.

Response: Thank you for this helpful suggestion. I have revised it. It is that “rapid population growth increases the domestic and agricultural demand for water resources. At the global scale, agriculture accounts for over 70% of all water withdrawn by the municipal, industrial (including energy) and agricultural sectors, and water shortage decreases agricultural productivity, affecting both incomes and food security [5].” (Line 38-Line 41, in ijerph-482170 - original.R1-Track Changes)

5. Chen, J.; Shi, H.Y.; Bellie, S.; Mervyn, R.P. Population, water, food, energy and dams. Renewable and Sustainable Energy Reviews. 2016, 56: 18-28.

AE5: Figure 1: the names are too small. Please revise.

Response:Thank you for this helpful suggestion. I have revised it. Please see Figure 1.

AE6:Figure 3: the label of the Y-axis is missing.

Response: Thank you for this helpful suggestion. I have revised it. Please see Figure 3

AE7: Figure 4: the label of the X-axis is missing.

Response: Thank you for this helpful suggestion. I have revised it. Please see Figure 4.

AE8: There are a few editing errors. I suggest the authors check the text carefully.

Response: Thank you for this helpful suggestion. I have checked the text carefully. I have used the English editing service of the MDPI to revise this manuscript (english-edited-9414).  I hope that I have corrected view of English grammar and writing style throughout the manuscript, and it is to your and the reviewers’ satisfaction.

Reviewer 2 Report

This reviewer made some hand-written suggestions/comments on the manuscript and scanned it.  Please, consider these comments for possible incoporation into the final version of your paper.

Author Response

Responses to Reviewer 2

AE1: This reviewer made some hand-written suggestions/comments on the manuscript and scanned it.  Please, consider these comments for possible incoporation into the final version of your paper.

Response: Thank you for this helpful suggestion. This manuscript has been revised carefully. I have corrected the problems that the reviewers brought forward point by point and have replied carefully, addressing the problems raised. And all revisions are clearly highlighted by using the "Track Changes" function in Microsoft Word 2016, so that changes are easily visible to you.  I have addressed your comments and revised your uploaded file carefully. I hope that I have modified and explained the manuscript clearly enough, and it is to your satisfaction. I have used the English editing service of the MDPI to revise this manuscript (english-edited-9414). We thank your comments which helped us improve the quality of this paper. 
